# Outcomes of kidney transplantation over a 16-year period in Korea: An analysis of the National Health Information Database

**Hyung Soon Lee**[1], **Minjin Kang**[2], **Banseok Kim**[3], **Yongjung Park**[3¤]*

**1** Department of Surgery, National Health Insurance Service Ilsan Hospital, Goyang, Republic of Korea,
**2** Research Institute, National Health Insurance Service Ilsan Hospital, Goyang, Republic of Korea,
**3** Department of Laboratory Medicine, National Health Insurance Service Ilsan Hospital, Goyang, Republic of Korea

¤ Current address: Department of Laboratory Medicine, Gangnam Severance Hospital, Yonsei University College of Medicine, Seoul, Republic of Korea

* ypark119@yuhs.ac

**Data Availability Statement:** Data cannot be shared publicly because the data on the National Health Insurance claim for all Korean residents are administered by the Korean National Health Insurance Service and are accessible only to

## Abstract

### Background

This study investigated the outcomes of kidney transplantation (KT) over a 16-year period in Korea and identified risk factors for graft failure using a nationwide population-based cohort.

### Methods

We investigated the Korean National Health Insurance Service-National Health Information Database. Health insurance claims for patients who underwent KT between 2002 and 2017 were analyzed.

### Results

The data from 18,331 patients who underwent their first KT were reviewed. The percentage of antithymocyte globulin (ATG) induction continuously increased from 2.0% in 2002 to 23.5% in 2017. Rituximab began to be used in 2008 and had increased to 141 patients (9.6%) in 2013. Acute rejection occurred in 17.3% of all patients in 2002 but decreased to 6.3% in 2017. The rejection-free survival rates were 78.8% at 6 months after KT, 76.1% after 1 year, 67.5% after 5 years, 61.7% after 10 years, and 56.7% after 15 years. The graft survival rates remained over 80% until 12 years after KT, and then rapidly decreased to 50.5% at 16 years after KT. In Cox's multivariate analysis, risk factors for graft failure included being male, more recent KT, KT from deceased donor, use of ATG, basiliximab, or rituximab, tacrolimus use as an initial calcineurin inhibitor, acute rejection history, and cytomegalovirus infection.

### Conclusions

ATG and rituximab use has gradually increased in Korea and more recent KT is associated with an increased risk of graft failure. Therefore, meticulous preoperative evaluation and

permitted users/researchers. Data are available from the National Health Insurance Sharing Service (https://nhiss.nhis.or.kr/) for researchers who meet the criteria for access to confidential data.

**Funding:** This study was supported by a grant from the National Health Insurance Service, Ilsan Hospital, Goyang, Republic of Korea. (Study no. 2019-20-002 and data no. REQ0000029924).

**Competing interests:** The authors have declared that no competing interests exist.

postoperative management are necessary in the case of recent KT with high risk of graft failure.

## Introduction

Kidney transplantation (KT) is considered the most valuable treatment option for patients with end stage renal disease (ESRD). KT offers better quality of life, cardiovascular stability, and improved survival compared to patients on dialysis [1]. The history of KT in Korea began with the first KT from a living donor in 1969 followed by the first deceased donor KT in 1979 [2]. However, the legal approval of brain death in Korea occurred relatively late, in 2000, and systematic discovery of brain death donors and the use of their organs has not been efficiently implemented yet. Thus, a shortage of donors continues to be a significant obstacle in Korea. Although living donor KT is actively being performed in Korea to overcome these donor shortages, the number of KTs being performed is far less than necessary and the waiting time grows longer [3].

Recent advances in immune suppression therapies and desensitization techniques enable KT regardless of ABO incompatibility or human leukocyte antigen (HLA) incompatibility (ABOi and HLAi, respectively) [4,5], and ABOi and HLAi KT have significantly increased the probability that patients with ESRD might receive living donor KT [6]. Although ABOi KT showed clinical outcomes comparable to ABO-compatible KT, ABOi KT required preoperative desensitization and the use of stronger immunosuppressants than ABO-compatible KT [7]. In this regard, ABOi KT may be a factor in increasing morbidity after KT [8]. On the other hand, HLAi KT still showed inferior clinical outcomes, in terms of acute rejection rates and allograft survival rates, compared with HLA compatible KT [9,10].

To date, although many publications have analyzed the outcome of KT worldwide, there is a lack of information regarding the nationwide long-term outcomes of KT patients. Large-scale data are needed to estimate long-term outcomes after KT, and to analyze associated risk factors of graft failure. Fortunately, South Korea has a unique single-insurer system: the National Health Insurance Service. The Korean National Health Insurance Service (KNHIS) is the only public medical insurance institution operated by the Ministry of Health and Welfare in Korea, and it provides health insurance services to nearly all Korean residents and contains large-scale medical information [11]. Thus, the aim of the current study was to investigate the outcomes of KT over a 16-year period in Korea using a KNHIS nationwide database, and to identify risk factors for graft failure after KT.

## Materials and methods

### Data source and study subject

All individuals in South Korea are obligated to join the KNHIS since the implementation of the service in 1989. The KNHIS controls all medical costs among individuals, health care providers, and the government. Therefore, medical data including personal information, diagnosis, medical treatment, and demographics of patients are centralized in the National Health Information Database (NHID) of the KNHIS. All insurance claims are exchanged through Electronic Data Interchange (EDI) codes. In addition, the KNHIS-NHID includes the diagnosis of patients by utilizing the Korean Classification of Diseases (KCD) codes, which is the Korean version of the International Classification of Diseases (ICD). We identified patients

who underwent KT between 2002 and 2017 from the KNHIS-NHID, by searching the EDI code for KT (R3280) during hospitalization.

## Variable definitions

We investigated the sex, age, type of donor (living or deceased), income quintiles, region of residence, type of hospital, length of hospital stay, type of dialysis, antithymocyte globulin (ATG)/basiliximab/rituximab use, type of immunosuppressant, type of steroid regimen, and development of acute rejection.

The length of hospital stay was defined as hospitalization for KT before and after surgery. The patients were divided into four groups based on age (<20, 20–39, 40–59 and ≥60 years). Meanwhile, the Organ Transplantation Law of South Korea requires a recipient to pay for the cost of donor nephrectomy in the case of living-donor KT, and the government shoulder the primary cost for the organ donation in the case of a deceased donor. Therefore, the donor type was able to be classified as a living donor when the EDI code for donor nephrectomy 'R3272' was charged to a recipient. The proportions of deceased- or living-donor KTs in this study were consistent with the statistics from the Korean Network for Organ Sharing (KONOS). Hospitals were classified into two types: general and tertiary. General hospitals were defined to have 100 or more beds. Tertiary hospitals were those designated by the Ministry of Health and Welfare based on requirements among general hospitals that specialize in highly-difficult medical practices for severe diseases. Currently, there are 42 tertiary hospitals in Korea. Since it is difficult to specify the exact time of kidney allograft rejection, due to the characteristics of the KNHIS claim data, acute rejection was defined as any case in which a diagnosis of kidney allograft rejection, by ICD-10 codes T86 and/or T86.1, was recorded during the KT-related hospitalization period. Recurrence of kidney allograft rejection during the postoperative follow-up period was defined as cases in which the rejection code was not recorded for more than 3 months, and those in which the rejection code was added again. Graft failure was defined as a KT recipient undergoing consistent dialyses for 3 months or longer during the follow-up period after the KT. In addition, we analyzed cytomegalovirus (CMV) infection and death during follow-up after KT. Because death certificates are automatically reported to the KNHIS, mortality was detected when healthcare coverage by the KNHIS was terminated.

## Statistical analysis

The Mann-Whitney U and Wilcoxon signed-rank tests were used to compare continuous variables between subject groups, and the Chi-square test was used to compare categorical variables. Graft survival and rejection-free survival were calculated using Kaplan–Meier survival analysis. Data were censored at the time of death, or at the last available follow-up. Cox's proportional hazard regression was performed to estimate the hazard ratios for various risk factors. All statistical analyses were performed by SAS 7.15 (SAS Institute Inc., Cary, NC, USA) and RStudio v1.1.463 (RStudio Inc., Boston, MA, USA), and $P$-values less than 0.05 were considered to be statistically significant.

## Ethics statement

This study was a retrospective cohort study and the study protocol was performed after approval from the Institutional Review Board (IRB) of National Health Insurance Service Ilsan Hospital (approval no. NHIMC 2019-01-001). Informed consent was waived by the IRB. All data were stored in the NHID server and fully anonymized before accessed by the researchers. The data in the server were analyzed with secure connection by a remote client computer in the Research Institute of Ilsan Hospital during August 2019 and January 2021. The results of

all analyses were taken out of the server after the data manager confirmed that personal information was not included in the result data.

In South Korea, all the transplants are carried out after being confirmed/approved by the KONOS that there is no ethical issue regarding the transplantation regardless of the living or deceased donors. None of the transplant donors was from a vulnerable population and all donors or next of kin provided written informed consent that was freely given.

## Results

### Patient selection and baseline characteristics

Within the KNHIS-NHID between 2002 and 2017, health insurance claims for a total of 20,750 patients during the KT hospitalization period were analyzed. Among them, 2,219 subjects who were considered as having undergone repeated KT (with a history of previous KT and/or with continuous immunosuppressive medication prescribed prior to KT) and 200 with unclear medical history were excluded from the analysis. The final cohort included a total of 18,331 KT patients over 16 years. Among a total of 18,331 KT recipients, 10,934 (59.6%) were male and 7,397 (40.4%) were female (Table 1). Living-donor KTs were performed in 11,398 (62.2%) patients. The number of KTs continued to increase from 509 in 2002 to 1,865 in 2017. Age at the time of KT most commonly fell in the 40- to 59-year-old group (10,655, 58.1%). Among the 18,331 KT cases, 15,662 (85.4%) were performed in tertiary hospitals. Compared

**Table 1. Characteristics of patients (n = 18,331).**

| Characteristics | | Donor type, N (%) | | | Total, N (%) |
|---|---|---|---|---|---|
| | | Living (n = 11,398) | Deceased (n = 6,933) | P-value | |
| Male | | 6,791 (59.6%) | 4,143 (59.8%) | 0.8127 | 10,934 (59.6%) |
| Age group (years) | < 20 | 288 (2.5%) | 155 (2.2%) | <0.0001 | 443 (2.4%) |
| | 20–39 | 2,621 (23.0%) | 1,927 (27.8%) | | 4,548 (24.8%) |
| | 40–59 | 6,648 (58.3%) | 4,007 (57.8%) | | 10,655 (58.1%) |
| | ≥ 60 | 1,841 (16.2%) | 844 (12.2%) | | 2,685 (14.6%) |
| Hospital type | Tertiary | 9,886 (86.7%) | 5,776 (83.3%) | <0.0001 | 15,662 (85.4%) |
| | General | 1,512 (13.3%) | 1,157 (16.7%) | | 2,669 (14.6%) |
| Year of surgery | 2002–2005 | 630 (5.5%) | 1,589 (22.9%) | <0.0001 | 2,219 (12.1%) |
| | 2006–2009 | 1,285 (11.3%) | 2,453 (35.4%) | | 3,738 (20.4%) |
| | 2010–2013 | 4,010 (35.2%) | 1,444 (20.8%) | | 5,454 (29.8%) |
| | 2014–2017 | 5,473 (48.0%) | 1,447 (20.9%) | | 6,920 (37.8%) |
| Length of hospital stay (days)[a] | | 26 (21–35) | 23 (18–30) | <0.0001 | 25 (20–33) |
| Perioperative use of | ATG | 1,603 (14.1%) | 715 (10.3%) | <0.0001 | 2,318 (12.6%) |
| | Basiliximab | 9,429 (82.7%) | 4,763 (68.7%) | <0.0001 | 14,192 (77.4%) |
| | Rituximab | 824 (7.2%) | 156 (2.3%) | <0.0001 | 980 (5.3%) |
| Initial calcineurin inhibitor | Cyclosporine | 1,325 (11.6%) | 2,068 (29.8%) | <0.0001 | 3,393 (18.5%) |
| | Tacrolimus | 9,434 (82.8%) | 4,263 (61.5%) | | 13,697 (74.7%) |
| | Other | 639 (5.6%) | 602 (8.7%) | | 1,241 (6.8%) |
| Steroid agent | Deflazacort | 1,406 (12.3%) | 595 (8.6%) | <0.0001 | 2,001 (10.9%) |
| | Dexa/betamethasone | 584 (5.1%) | 285 (4.1%) | | 869 (4.7%) |
| | Fludro/hydrocortisone | 1,336 (11.7%) | 587 (8.5%) | | 1,923 (10.5%) |
| | (Methyl)prednisolone | 8,072 (70.8%) | 5,465 (78.8%) | | 13,537 (73.8%) |

ATG, antithymocyte globulin.

[a]Values are median and 1st to 3rd quartiles.

with the patients that received deceased-donor KT, living-donor KT cases showed higher proportions of older age groups, tertiary hospital as an institution of operation, and more recent KT cases. Living-donor KT cases also exhibited longer hospital stay and higher proportions of cases using ATG, basiliximab, rituximab, tacrolimus, and deflazacort.

## Induction therapy and desensitization agents

The number of patients using ATG continued to increase from 10 (2.0%) in 2002 to 438 (23.5%) in 2017 (Table 2), and the proportion of patients who used basiliximab increased from 28.9% in 2002 to 94.0% in 2010, and then decreased to 78.7% in 2017. Rituximab began to be used in 2008 and increased to 141 patients (9.6%) in 2013. However, the use of rituximab declined again to 117 (6.3%) patients in 2017.

## Initial immunosuppressive regimen

The combination of tacrolimus and mycophenolate mofetil (MMF) was most commonly prescribed as an initial immunosuppressive regimen in 11,979 (65.3%) of the 18,331 KT patients followed by cyclosporine and MMF in 2,918 (15.9%). The combination of tacrolimus and MMF was used in 116 patients (22.8%) in 2002 and its rate of use increased gradually to 1,865 patients (89.3%) in 2017. On the other hand, cyclosporine and MMF combinations were used in 260 patients (51.1%) in 2002 but only in 24 patients (1.3%) in 2017. Overall, the use of tacrolimus increased steadily while the use of cyclosporine continued to decrease and the rates of use of these two treatments was reversed between 2006 and 2007. With regard to steroid agents, (methyl)prednisolone accounted for 73.8% of the steroids used during the entire study period, with 10.9% being deflazacort and 10.5% being fludrocortisone/hydrocortisone. In particular, deflazacort was used in 3.3% of patients in 2002, but its usage increased to 16.1% in 2010, and then decreased again to 10.2% in 2017.

## Acute rejection

In 2002, 17.3% of patients had acute rejection during the hospital stay for KT. However, acute rejection during the hospital stay for KT decreased with time, and only 6.3% had acute rejection in 2017. Although the incidence of acute rejection was 13.2% in 2007, it remained between 5.1% and 8.8% each year after a sharp decline to 8.3% in 2008 (Fig 1). Among the patients who received KT in 2002, 64.2% of the patients had a rejection during the follow-up period, and 12.4% of the patients who undergone KT in 2017 had a rejection. The rejection-free survival rates for living-donor KT cases were 82.6% at 6 months after KT, 80.3% after 1 year, 71.8%

**Table 2. Induction therapy and desensitization agents.**

| Agent | Year of kidney transplant | | | | | | | | | | | | | | | | Total |
|---|---|---|---|---|---|---|---|---|---|---|---|---|---|---|---|---|---|
| | 2002 | 2003 | 2004 | 2005 | 2006 | 2007 | 2008 | 2009 | 2010 | 2011 | 2012 | 2013 | 2014 | 2015 | 2016 | 2017 | |
| ATG | 10 | 23 | 27 | 17 | 51 | 71 | 73 | 45 | 65 | 116 | 155 | 222 | 279 | 316 | 410 | 438 | 2,318 |
| (%) | 2.0% | 3.9% | 4.6% | 3.2% | 6.1% | 8.8% | 7.2% | 4.1% | 5.8% | 8.3% | 10.4% | 15.2% | 18.1% | 19.8% | 21.4% | 23.5% | 12.6% |
| Basiliximab | 147 | 199 | 187 | 184 | 515 | 541 | 893 | 957 | 1,046 | 1,269 | 1,346 | 1,279 | 1,280 | 1,327 | 1,554 | 1,468 | 14,192 |
| (%) | 28.9% | 33.4% | 31.9% | 34.8% | 62.0% | 66.8% | 88.7% | 87.7% | 94.0% | 91.2% | 90.5% | 87.4% | 83.2% | 82.9% | 81.1% | 78.7% | 77.4% |
| Rituximab | | | | | | | 3 | 36 | 70 | 105 | 129 | 141 | 135 | 121 | 123 | 117 | 980 |
| (%) | | | | | | | 0.3% | 3.3% | 6.3% | 7.5% | 8.7% | 9.6% | 8.8% | 7.6% | 6.4% | 6.3% | 5.3% |
| Total | 509 | 595 | 586 | 529 | 830 | 810 | 1,007 | 1,091 | 1,113 | 1,391 | 1,487 | 1,463 | 1,539 | 1,600 | 1,916 | 1,865 | 18,331 |

ATG, antithymocyte globulin.

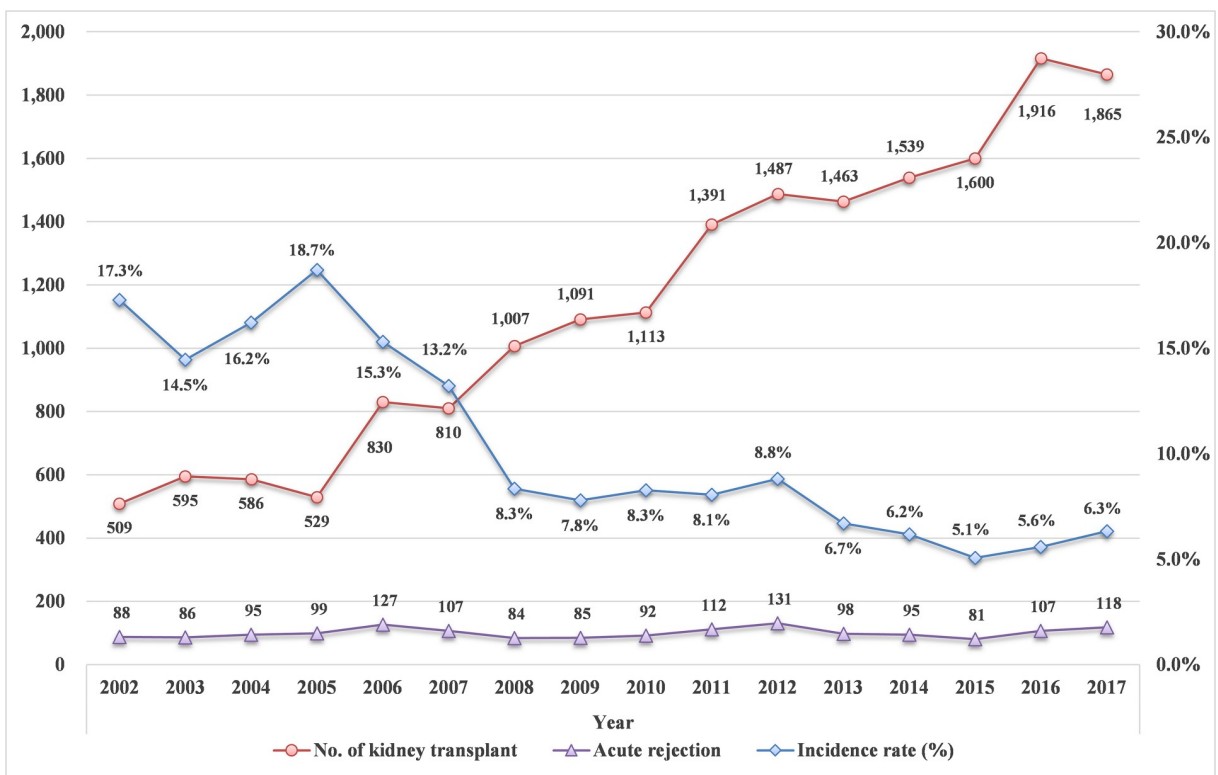

**Fig 1. The incidence of acute rejection during hospital stay for kidney transplant by year.** The incidence rate of acute rejection shows a tendency of steady decrease over 16 years.

after 5 years, 65.7% after 10 years, and 60.9% after 14 years, and those for deceased-donor KTs were 73.7% at 6 months after KT, 70.2% after 1 year, 61.6% after 5 years, 56.1% after 10 years, and 53.3% after 14 years (Fig 2). Patient characteristics were stratified according to the occurrence of acute rejection during hospital stay for KT and are summarized in Table 3. In the patients with acute rejection, the proportion of female patients, 20- and 59-year-old age group, and the deceased-donor KT was higher than those in the KT recipients without acute rejection. There was no significant difference in hospital type. However, the proportion of patients who underwent KT between 2002 and 2009 was significantly higher in the acute rejection group, and hospital stays were also significantly longer in the acute rejection group. The proportion of patients receiving ATG or rituximab were also significantly higher in patients with acute rejection, whereas those receiving basiliximab was significantly lower in patients with acute rejection. Additionally, the proportion of patients receiving cyclosporine, compared to tacrolimus, as an initial immunosuppressant was also significantly higher in patients with acute rejection. With regard to steroid agents, the proportion of patients who received (methyl) prednisolone was significantly higher for patients without acute rejection, while the proportion of patients who received either dexa/betamethasone or fludro/hydrocortisone was significantly higher for patients with acute rejection.

## Graft failure

Graft failure occurred in 7.5% of patients during the entire follow-up period. The graft survival rates for living-donor KT cases were 99.8% after one year from KT, 98.0% after 5 years, 90.3% after 10 years, and 71.3% after 15 years, and those for deceased-donor KTs were 99.7% after

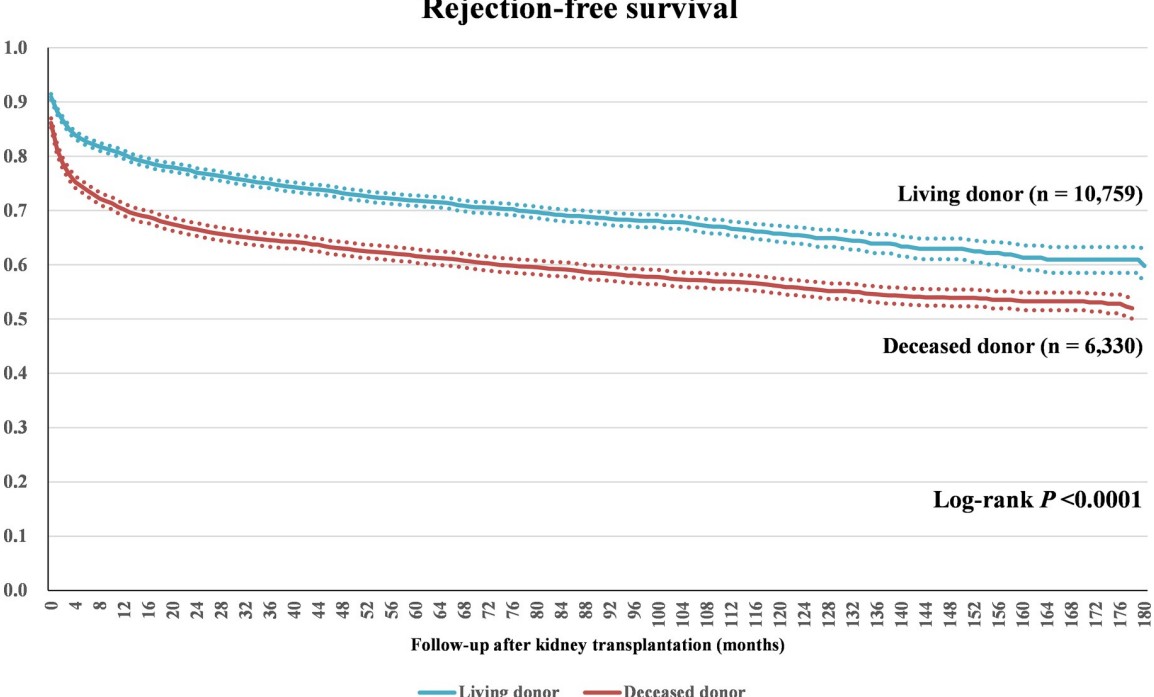

**Fig 2. Rejection-free survival after kidney transplant.** Dashed lines demonstrate 95% confidence intervals of the rejection-free survival rate.

one year, 96.7% after 5 years, 85.6% after 10 years, and 66.5% after 15 years (Fig 3A). In addition, the graft survival rates for cases undergone KT between the years 2002 and 2009 were 99.9% after one year, 99.2% after 5 years, and 98.3% after 8 years, and those between the years 2010 and 2017 were 99.6% after one year, 95.6% after 5 years, and 70.4% after 8 years (Fig 3B). Table 4 summarizes the differences in patient characteristics, predisposing factors, and types of initial immunosuppressant used according to the occurrence of graft failure. The proportions of male patients, recipients aged 20 to 39 years old at the time of KT, and the deceased-donor KT were significantly higher in the patients with graft failure, and that of patients who received basiliximab during the hospital stay for KT was significantly higher in patients without graft failure. Among patients with graft failure, a significantly higher proportion received cyclosporine as an initial calcineurin inhibitor, compared to tacrolimus. Acute rejection during the hospital stay for KT and CMV infection prior to graft failure was significantly higher in patients with graft failure.

### Risk factor of graft failure

Male sex, more recent KT, and deceased-donor KT were significant independent risk factors for graft failure (Table 5). The use of ATG, basiliximab, and rituximab, and use of tacrolimus as an initial calcineurin inhibitor, and deflazacort during hospital stay for KT were also associated with graft failure. Furthermore, CMV infection prior to graft failure and acute rejection during the hospital stay for KT were independent risk factors for graft failure.

### Discussion

The present study demonstrates that ATG induction and the use of rituximab have steadily increased in Korea. Acute rejection occurred in 17.3% of all patients in 2002 but decreased to

**Table 3. Characteristics of patient according to the development of acute rejection during the hospital stay for kidney transplant.**

| Characteristics | | | Without acute rejection (n = 16,726) | Acute rejection (n = 1,605) | *P*-value |
|---|---|---|---|---|---|
| **Male** | | | 10,080 (60.3%) | 854 (53.2%) | <0.0001 |
| **Age (year)**[a] | | | 46.8 ± 12.4 | 46.3 ± 12.0 | 0.0934 |
| **Age group (years)** | | < 20 | 413 (2.5%) | 30 (1.9%) | 0.0293 |
| | | 20–39 | 4,129 (24.7%) | 419 (26.1%) | |
| | | 40–59 | 9,701 (58.0%) | 954 (59.4%) | |
| | | ≥ 60 | 2,483 (14.8%) | 202 (12.6%) | |
| **Income level** | | Lower 50% | 15,767 (94.3%) | 1,498 (93.3%) | 0.1271 |
| | | Upper 50% | 959 (5.7%) | 107 (6.7%) | |
| **Hospital type** | | Tertiary | 14,317 (85.6%) | 1,345 (83.8%) | 0.0513 |
| | | General | 2,409 (14.4%) | 260 (16.2%) | |
| **Year of surgery** | | 2002–2005 | 1,851 (11.1%) | 368 (22.9%) | <0.0001 |
| | | 2006–2009 | 3,335 (19.9%) | 403 (25.1%) | |
| | | 2010–2013 | 5,021 (30.0%) | 433 (27.0%) | |
| | | 2014–2017 | 6,519 (39.0%) | 401 (25.0%) | |
| **Length of hospital stay (days)**[b] | | | 25 (19–33) | 30 (22–43) | <0.0001[b] |
| **Donor type** | | Living | 10,546 (63.1%) | 852 (53.1%) | <0.0001 |
| | | Deceased | 6,180 (36.9%) | 753 (46.9%) | |
| **Perioperative use of** | | ATG | 1,908 (11.4%) | 410 (25.5%) | <0.0001 |
| | | Basiliximab | 13,065 (78.1%) | 1,127 (70.2%) | <0.0001 |
| | | Rituximab | 780 (4.7%) | 200 (12.5%) | <0.0001 |
| **Initial calcineurin inhibitor** | | Cyclosporine | 3,719 (22.2%) | 512 (31.9%) | <0.0001 |
| | | Tacrolimus | 12,642 (75.6%) | 1,055 (65.7%) | |
| | | Other | 365 (2.2%) | 38 (2.4%) | |
| **Initial steroid agent** | | Deflazacort | 1,825 (10.9%) | 176 (11.0%) | <0.0001 |
| | | Dexa/betamethasone | 729 (4.4%) | 140 (8.7%) | |
| | | Fludro/hydrocortisone | 1,617 (9.7%) | 306 (19.1%) | |
| | | (Methyl)prednisolone | 12,554 (75.1%) | 983 (61.2%) | |

Values are indicated as number (percentage) unless indicated otherwise. ATG, antithymocyte globulin.

[a]Values are mean ± standard deviation.

[b]Values are median and 1st to 3rd quartiles.

6.3% in 2017. The living-donor KTs showed significantly higher rejection-free survival rates when compared with deceased-donor KTs. Living-donor KTs also exhibited significantly higher graft survival than deceased-donor KTs. Particularly, KTs performed between the years 2002 and 2009 showed significantly higher graft survival than KTs carried out between the years 2010 and 2017. Male sex, more recent operation, deceased-donor KT, use of ATG, basiliximab, rituximab, and tacrolimus as an initial calcineurin inhibitor, and deflazacort during hospital stay for KT were associated with graft failure. CMV infection and acute rejection history were also determined to be risk factors for graft failure. These results suggest that the long-term outcomes of KT in Korea are good and that, in recent times, KT in Korea tends to be performed in patients with high risk of graft failure.

KT in patients with high risk of graft failure is a growing phenomenon due to the shortage of donors and the development of various desensitization protocols. However, KT in immunologically high-risk recipients have affected both patient and graft survival [8,12]. ABOi and HLAi are the major immunological barriers in KT. Various desensitization protocols to overcome the ABOi and HLA barriers, such as donor-specific HLA antibodies, have increased the

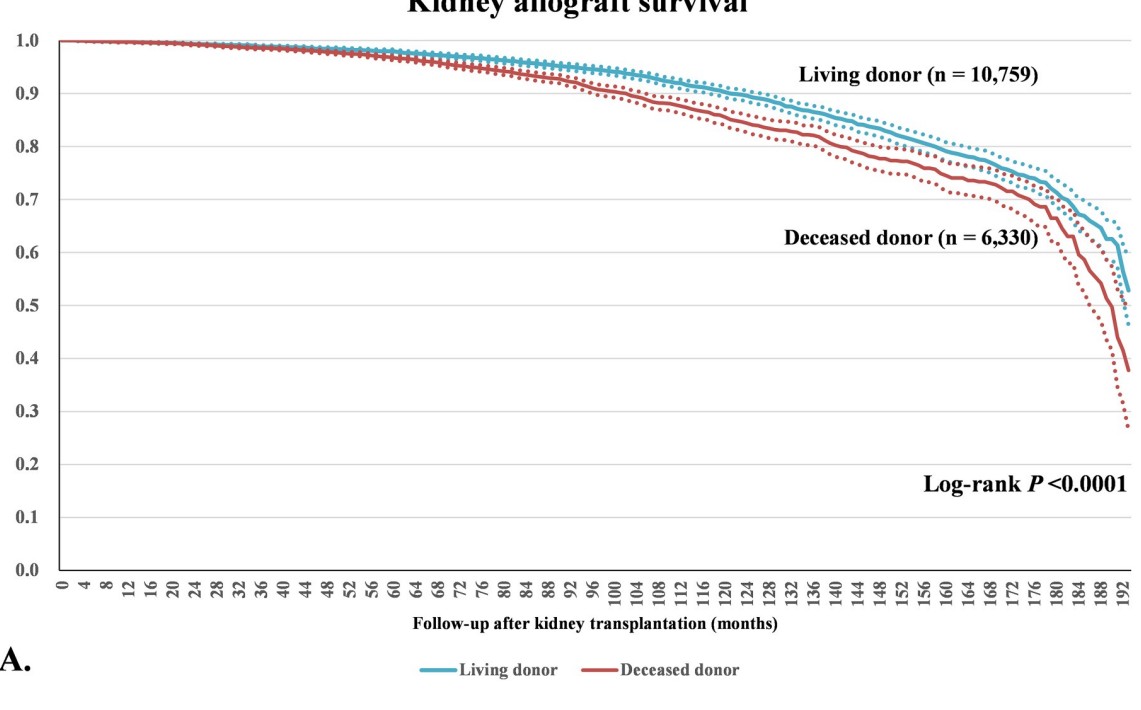

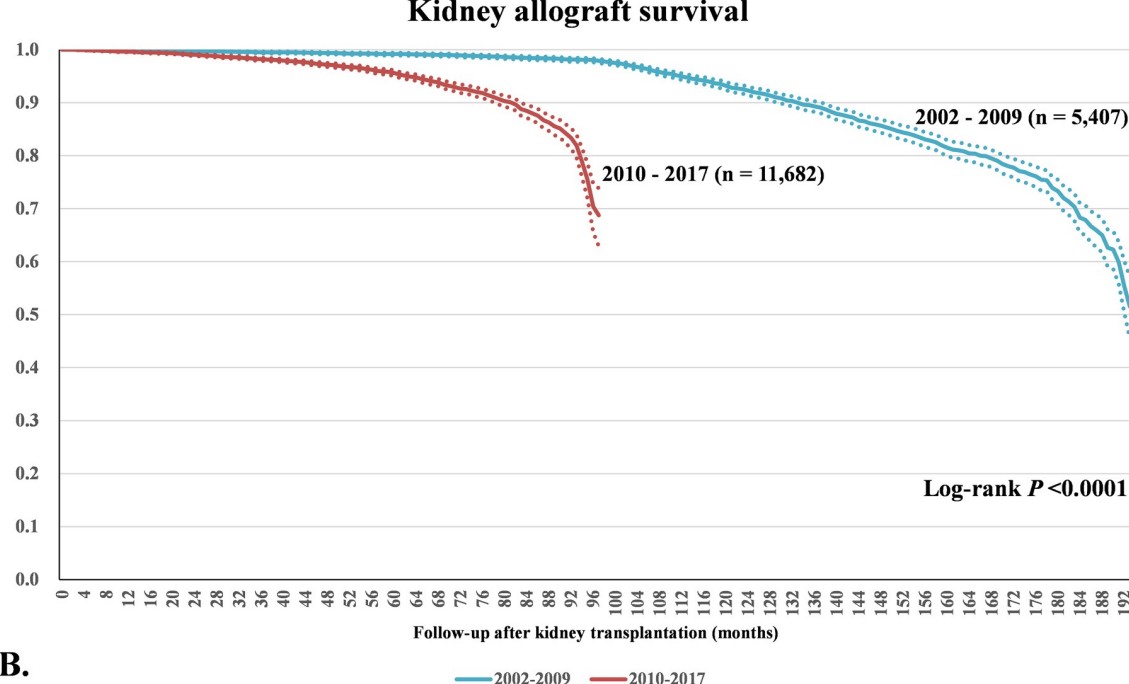

**Fig 3. Graft survival after kidney transplant over 16 years.** Dashed lines demonstrate 95% confidence intervals of the graft survival rate.

opportunity for KT to be performed in increasing numbers of patients [13]. The numbers of transplant centers and cases of KT after desensitization are increasing in Korea [5,14]. Despite a low incidence of hyperacute rejection and acceptable short-term outcomes in those patients,

Table 4. Characteristics of patient according to the development of graft failure.

| Characteristics | | Without graft failure (n = 15,850) | Graft failure (n = 1,239) | P-value |
|---|---|---|---|---|
| **Male** | | 9,386 (59.2%) | 772 (62.3%) | 0.0329 |
| **Age (years)[a]** | | 47.0 ± 12.3 | 43.8 ± 12.4 | <0.0001 |
| **Age group (years)** | **< 20** | 381 (2.4%) | 37, 3.0% | <0.0001 |
| | **20–39** | 3,794 (23.9%) | 420 (33.9%) | |
| | **40–59** | 9,261 (58.4%) | 655 (52.9%) | |
| | **≥ 60** | 2,414 (15.2%) | 127 (10.3%) | |
| **Income level** | **Lower 50%** | 14,920 (94.1%) | 1,170 (94.4%) | 0.6662 |
| | **Upper 50%** | 930 (5.9%) | 69 (5.6%) | |
| **Hospital type** | **Tertiary** | 13,522 (85.3%) | 1,049 (84.7%) | 0.5359 |
| | **General** | 2,328 (14.7%) | 190 (15.3%) | |
| **Year of surgery** | **2002–2005** | 1,716 (10.8%) | 313 (25.3%) | <0.0001 |
| | **2006–2009** | 2,971 (18.7%) | 407 (32.8%) | |
| | **2010–2013** | 4,749 (30.0%) | 356 (28.7%) | |
| | **2014–2017** | 6,414 (40.5%) | 163 (13.2%) | |
| **Donor type** | **Living** | 10,182 (64.2%) | 577 (46.6%) | <0.0001 |
| | **Deceased** | 5,668 (35.8%) | 662 (53.4%) | |
| **Perioperative use of** | **ATG** | 2,037 (12.9%) | 136 (11.0%) | 0.0564 |
| | **Basiliximab** | 12,376 (78.1%) | 898 (72.5%) | <0.0001 |
| | **Rituximab** | 865 (5.5%) | 64 (5.2%) | 0.6625 |
| **Initial calcineurin inhibitor** | **Cyclosporine** | 3,000 (18.9%) | 393 (31.7%) | <0.0001 |
| | **Tacrolimus** | 12,850 (81.1%) | 846 (68.3%) | |
| **Initial steroid agent** | **Deflazacort** | 1,690 (10.7%) | 132 (10.7%) | 0.0732 |
| | **Dexa/betamethasone** | 748 (4.7%) | 66 (5.3%) | |
| | **Fludro/hydrocortisone** | 1,599 (10.1%) | 151 (12.2%) | |
| | **(Methyl)prednisolone** | 11,813 (74.5%) | 890 (71.8%) | |
| **Acute rejection[b]** | | 1,177 (7.4%) | 221 (17.8%) | <0.0001 |
| **CMV infection[c]** | | 2,299 (14.5%) | 303 (24.5%) | <0.0001 |

Values are indicated as number (percentage) unless indicated otherwise. ATG, antithymocyte globulin; CMV, cytomegalovirus.

[a]Values are mean ± standard deviation.

[b]Acute rejection during hospitalization for kidney transplantation.

[c]CMV infection at the time prior to graft failure.

antibody-mediated rejection remains a significant challenge even after successful desensitization. Acute rejection episodes after KT are considered a risk factor for short-term and long-term graft survival [15]. The incidence of acute rejection varies worldwide, depending on the diagnostic criteria used, recipient sensitization, and the immunosuppressive regimen, and ranges from 3.1% to as high as 30% to 40% [16,17]. In the current study, the incidence of acute rejection during the hospitalization period and within the year of KT surgery steadily decreased from 17.3% in 2002 to 6.3% in 2017, which is a relatively low incidence of acute rejection. This may be caused by the definition of acute rejection in our data as that occurring during the hospitalization period related to KT. Also, the current study showed that the incidence of acute rejection had declined over the year with a sharp decline to 8.3% in 2008 (Fig 1). These results may be associated with the increased perioperative use of desensitizing agents (Table 2).

The use of ATG as an induction immunosuppressive agent has been increasing in recent years, replacing basiliximab therapy [18]. In Korea, the number of deceased-donor KT and the

**Table 5. Multivariate Cox-proportional hazard model for risk factor of graft failure.**

| Variables | Hazard ratio | 95% confidence interval | *P*-value |
|---|---|---|---|
| **Male** | 1.235 | (1.099–1.387) | 0.0004 |
| **Age group (years)** | | | |
| < 20 | 1.000 | | |
| 20–39 | 1.188 | (0.843–1.674) | 0.3248 |
| 40–59 | 1.027 | (0.732–1.441) | 0.8774 |
| ≥ 60 | 1.373 | (0.944–1.996) | 0.0970 |
| **Year of surgery** | | | |
| 2002–2009 | 1.000 | | |
| 2010–2017 | 9.732 | (7.632–12.410) | <0.0001 |
| **Hospital type** | | | |
| Tertiary | 1.000 | | |
| General | 0.885 | (0.755–1.039) | 0.1351 |
| **Donor type** | | | |
| Living | 1.000 | | |
| Deceased | 1.160 | (1.025–1.312) | 0.0190 |
| **ATG use** | 2.074 | (1.657–2.596) | <0.0001 |
| **Basiliximab use** | 1.945 | (1.678–2.255) | <0.0001 |
| **Rituximab use** | 1.339 | (1.021–1.756) | 0.0346 |
| **Initial calcineurin inhibitor** | | | |
| Cyclosporine | 1.000 | | |
| Tacrolimus | 1.375 | (1.198–1.576) | <0.0001 |
| **Initial steroid agent** | | | |
| Deflazacort | 1.000 | | |
| Dexa/betamethasone | 0.911 | (0.674–1.230) | 0.5424 |
| Fludro/hydrocortisone | 1.026 | (0.796–1.321) | 0.8438 |
| (Methyl)prednisolone | 0.799 | (0.660–0.966) | 0.0208 |
| **CMV infection** | 1.638 | (1.435–1.868) | <0.0001 |
| **Acute rejection** | 1.598 | (1.372–1.863) | <0.0001 |

ATG, antithymocyte globulin; CMV, cytomegalovirus; KT, kidney transplantation.

use of expanded donor criteria have increased [2,19,20]. Furthermore, several transplantation centers in Korea have recently begun using ATG induction therapy in KTs from deceased donors with acute kidney injury in the expectation of better graft outcomes [21,22]. These trends are also observed in a multicenter study conducted in Korea in 2017. Chang *et al.* demonstrated that ATG was more frequently used in highly sensitized patients, deceased-donor transplants, re-transplants, and ABO incompatible transplants [23]. In accordance with the previous study, basiliximab usage in our data continued to increase from 28.9% in 2002 to 94.0% in 2010, and then decreased. On the other hand, ATG use increased every year. Therefore, there may be a current trend of high-risk patients undergoing KT after ATG administration to expand the donor pool in Korea. Meanwhile, regimens used for induction therapies were associated with graft failure in our data. In cohort studies, the temporal relationship between variables and outcomes does not always imply a direct causal relationship. Therefore, attention should be paid to the interpretation of variables for causality. ATG and rituximab are generally used for induction or desensitization therapy in recipients with higher immunological risk of graft rejection but also can be used to treat acute rejection. Thus, it would be desirable to interpret that the use of these drugs itself is not a direct risk factor for graft failure, but

the drugs were used in high-risk recipients for treating acute rejection as well as for induction therapy, which resulted in the association between the use of these drugs and the outcome of graft failure.

Immunosuppressive maintenance therapy differs in the types and combinations of drugs used in the transplant center. The current study showed that the use of cyclosporine in KT recipients steadily decreased, whereas the use of tacrolimus steadily increased. According to previous reports, cyclosporine has been used since the 1980s, but low-dose tacrolimus/MMF has been shown to result in higher 1-year graft survival and less acute rejection than the cyclosporine/MMF combination [24]. Therefore, these trends may cause a shift in the major immunosuppressive agent used in Korea, from cyclosporine to tacrolimus. Deflazacort is a heterocyclic corticosteroid, and is characterized by high efficacy and good tolerability because of its substantial lack of fluid retention and low interference with carbohydrate and phospho-calcium metabolism [25]. Due to its excellent anti-inflammatory properties and good tolerability, deflazacort is preferred in patients with osteoporosis or diabetes [26]. However, there is still debate on whether the immunosuppressive activity and side effects of deflazacort are comparable to those of (methyl)prednisolone [27,28]. In the current study, deflazacort was used in 3.3% of patients in 2002, but increased in use for 16.1% of patients in 2010, and then decreased again in 10.2% of patients in 2017. As compared with (methyl)prednisolone, the use of deflazacort during hospital stay for KT was a significant independent risk factor for graft failure. Therefore, the use of deflazacort during hospital stay for KT requires more attention for graft function.

This study has several limitations. First, because we analyzed data from the database of national health insurance claims, laboratory, imaging, and pathological results, lifestyle factors or details of donor characteristics including ABO and HLA compatibilities were not able to be included in our data. Second, we focused only on residents of South Korea, and thus, the findings of our study would be hard to be generalized to other countries or ethnic groups. Third, we could not identify non-reimbursed drugs or medical practices for KT.

In summary, the current study demonstrated that the long-term outcomes of KT in Korea were good. However, ATG and rituximab use has steadily increased in Korea, and more recent KT was associated with an increased risk of graft failure. Although analyses on the HLA mismatch, ABO compatibility, pre-transplant sensitization level, and desensitization protocol were not included in this study, our results suggests that the recent KT might be performed in large number of patients with high risk of rejection or graft failure. Therefore, meticulous preoperative evaluation and postoperative management are necessary for recent KT in Korea, especially for patients with high risk of rejection or graft failure. Additionally, the risk factors affecting KT outcomes in Korea needs to be further investigated in well-designed large-scale studies.

## Author Contributions

**Conceptualization:** Hyung Soon Lee, Yongjung Park.

**Data curation:** Minjin Kang.

**Formal analysis:** Minjin Kang, Banseok Kim.

**Funding acquisition:** Yongjung Park.

**Investigation:** Minjin Kang, Yongjung Park.

**Methodology:** Hyung Soon Lee, Banseok Kim, Yongjung Park.

**Software:** Minjin Kang.

**Supervision:** Yongjung Park.

**Validation:** Hyung Soon Lee, Banseok Kim.

**Visualization:** Hyung Soon Lee, Minjin Kang, Banseok Kim.

**Writing – original draft:** Hyung Soon Lee.

**Writing – review & editing:** Yongjung Park.

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
