## [Decision Letter · Decision Letter 0]

6 Jan 2021

PONE-D-20-37168

Outcomes of Kidney Transplantation Over a 16-year Period in Korea: An Analysis of the National Health Information Database

PLOS ONE

Dear Dr. Park,

Thank you for submitting your manuscript to PLOS ONE. After careful consideration, we feel that it has merit but does not fully meet PLOS ONE’s publication criteria as it currently stands. Therefore, we invite you to submit a revised version of the manuscript that addresses the points raised during the review process.

The manuscript is the summary in the detail analysis of transplant outcomes in Korea during the last sixteen years. The study include data about: patient characteristics; immunosuppression therapy; incidence of graft rejection and graft loss. Although, it is a useful review about the kidney transplantation system in Korea, the authors need to address all specifically indicated aspects by the reviewers.

The authors need to show the type of donors. It is also necessary to clearly show the results with deceased donors vs. live donors. Cosequently, tables 1, 3 and 4 need to be modified to clearly show the results for deceased donors and for live donors. In addition, the appropiate statistical analysis needs to be performed. All these changes will require significant revisions to the manuscript.

The authors need to provide enough information supported by the appropiate statistical analysis to make sure that there is not an over-interpratation of their results. 

The authors need to address specific comments by the second reviewer: ABO compatibility (whether or not ABOi transplants were included), and pre-transplant sensitization levels. 

The conclusions need to be re-evaluated regarding risk factors impacting transplantation outcomes in Korea. 

Very careful ststistical analysis for graft failure risk needs to be performed using a Cox regression model, testing for proportional hazard distribution with variables.

In response to the reviewer # 1:

The presented data are not robust, and not in-depth. Additional data need to be included divided into deceased and living donors.

The results need to reflect the standard reporting of transplant data accepted universally for example 'survival free rejection' is no commonly used.  Overall survival vs graft survival, % acute rejection (report Tcell vs AMR vs Mixed), what % was deceased donor vs Living Donor kidney tx., HLA match, ABO vs ABOi, Desensitization protocol used, ATG dosing?.  

It is particularly interesting and counter to accepted results that ATG use led to worse outcomes. The authors do not explain this finding. The thesis was mixed, a more descriptive manuscript would suffice with the data and findings clearly presented.

In response to Reviewer # 2:

This neatly organized and well written study summarizes the analyses of transplant outcomes in Korea over the last sixteen years. The manuscript is a useful presentation of transplantation experience in this country. Results on patient characteristics, immunosuppression therapy use and as wells graft rejection and loss statistics are presented. Overall, it is a good manuscript for readers wishing to learn about kidney transplantation system in Korea. Some important details were missed, however.
Most importantly, please kindly indicate the donor type in the patient cohort. Were all patients included in the study a deceased donor transplant recipients, or it was a mix of deceased and living donor transplant recipients. If the latter is the case, donor type (and donor relationship for living donors) should be shown in Tables 1, 3 and 4, and, importantly, included in Cox regression model as a variable and shown in Table 5. This would constitute a major revision to the manuscript.

Overall, I would caution against over-interpreting the results of variable distribution comparison between patients with or without acute rejection and graft loss, and Cox regression, due to the lack of information on donor type, ABO compatibility (whether or not ABOi transplants were included), pre-transplant sensitization level. The conclusions of the paper regarding risk factors impacting transplantation outcomes in Korea should be at least paraphrased to reflect these limitations. In addition, for a truly vigorous statistical testing of graft failure risk in a Cox regression model, testing for proportional hazard distribution of included variables would be beneficial, though I consider it optional for the scope of this manuscript.

We look forward to receiving your revised manuscript.

Kind regards,

Stanislaw Stepkowski

Academic Editor

PLOS ONE

Journal Requirements:

2. In the ethics statement in the manuscript and in the online submission form, please provide additional information about the patient records/samples used in your retrospective study, including:

a) whether all data were fully anonymized before you accessed them;

b) the date range (month and year) during which patients' medical records/samples were accessed.

3. We note that your study involved tissue/organ transplantation. Please provide the following information regarding tissue/organ donors for transplantation cases analyzed in your study.

a. Please provide the source(s) of the transplanted tissue/organs used in the study, including the institution name and a non-identifying description of the donor(s).

b. Please state in your response letter and ethics statement whether the transplant cases for this study involved any vulnerable populations; for example, tissue/organs from prisoners, subjects with reduced mental capacity due to illness or age, or minors.

- If a vulnerable population was used, please describe the population, justify the decision to use tissue/organ donations from this group, and clearly describe what measures were taken in the informed consent procedure to assure protection of the vulnerable group and avoid coercion.

- If a vulnerable population was not used, please state in your ethics statement, “None of the transplant donors was from a vulnerable population and all donors or next of kin provided written informed consent that was freely given.”

c. In the Methods, please provide detailed information about the procedure by which informed consent was obtained from organ/tissue donors or their next of kin. In addition, please provide a blank example of the form used to obtain consent from donors, and an English translation if the original is in a different language.

d. Please indicate whether the donors were previously registered as organ donors. If tissues/organs were obtained from deceased donors or cadavers, please provide details as to the donors’ cause(s) of death.

e. Please provide the participant recruitment dates and the period during which transplant procedures were done (as month and year).

f. Please discuss whether medical costs were covered or other cash payments were provided to the family of the donor. If so, please specify the value of this support (in local currency and equivalent to U.S. dollars)."

4. Thank you for stating the following in the Title page of your manuscript:

'Funding resource: This study was supported by a grant from the National Health Insurance Service, Ilsan Hospital, Goyang, Republic of Korea. (Study no. 2019-20-002 and data no. REQ0000029924).'

a. Please remove any funding-related text from the manuscript and let us know how you would like to update your Funding Statement. Currently, your Funding Statement reads as follows: 'NO'

Additional Editor Comments:

The manuscript is the summary in the detail analysis of transplant outcomes in Korea during the last sixteen years. The study include data about: patient characteristics; immunosuppression therapy; incidence of graft rejection and graft loss. Although, it is a useful review about the kidney transplantation system in Korea, the authors need to address all specifically indicated aspects by the reviewers.

The authors need to show the type of donors. It is also necessary to clearly show the results with deceased donors vs. live donors. Cosequently, tables 1, 3 and 4 need to be modified to clearly show the results for deceased donors and for live donors. In addition, the appropiate statistical analysis needs to be performed. All these changes will require significant revisions to the manuscript.

The authors need to provide enough information supported by the appropiate statistical analysis to make sure that there is not an over-interpratation of their results.

The authors need to address specific comments by the second reviewer: ABO compatibility (whether or not ABOi transplants were included), and pre-transplant sensitization levels.

The conclusions need to be re-evaluated regarding risk factors impacting transplantation outcomes in Korea.

Very careful ststistical analysis for graft failure risk needs to be performed using a Cox regression model, testing for proportional hazard distribution with variables.

In response to the reviewer # 1:

The presented data are not robust, and not in-depth. Additional data need to be included divided into deceased and living donors.

The results need to reflect the standard reporting of transplant data accepted universally for example 'survival free rejection' is no commonly used. Overall survival vs graft survival, % acute rejection (report Tcell vs AMR vs Mixed), what % was deceased donor vs Living Donor kidney tx., HLA match, ABO vs ABOi, Desensitization protocol used, ATG dosing?.

It is particularly interesting and counter to accepted results that ATG use led to worse outcomes. The authors do not explain this finding. The thesis was mixed, a more descriptive manuscript would suffice with the data and findings clearly presented.

In response to Reviewer # 2:

This neatly organized and well written study summarizes the analyses of transplant outcomes in Korea over the last sixteen years. The manuscript is a useful presentation of transplantation experience in this country. Results on patient characteristics, immunosuppression therapy use and as wells graft rejection and loss statistics are presented. Overall, it is a good manuscript for readers wishing to learn about kidney transplantation system in Korea. Some important details were missed, however.

Most importantly, please kindly indicate the donor type in the patient cohort. Were all patients included in the study a deceased donor transplant recipients, or it was a mix of deceased and living donor transplant recipients. If the latter is the case, donor type (and donor relationship for living donors) should be shown in Tables 1, 3 and 4, and, importantly, included in Cox regression model as a variable and shown in Table 5. This would constitute a major revision to the manuscript.

Overall, I would caution against over-interpreting the results of variable distribution comparison between patients with or without acute rejection and graft loss, and Cox regression, due to the lack of information on donor type, ABO compatibility (whether or not ABOi transplants were included), pre-transplant sensitization level. The conclusions of the paper regarding risk factors impacting transplantation outcomes in Korea should be at least paraphrased to reflect these limitations. In addition, for a truly vigorous statistical testing of graft failure risk in a Cox regression model, testing for proportional hazard distribution of included variables would be beneficial, though I consider it optional for the scope of this manuscript.

A detailed table of results and a KM curve to present the results will be important. The discussion of the manuscript are centered around results not presente

Reviewers' comments:

Reviewer's Responses to Questions

**Comments to the Author**

1. Is the manuscript technically sound, and do the data support the conclusions?

Reviewer #1: Yes

Reviewer #2: No

2. Has the statistical analysis been performed appropriately and rigorously? 

Reviewer #1: Yes

Reviewer #2: No

3. Have the authors made all data underlying the findings in their manuscript fully available?

Reviewer #1: Yes

Reviewer #2: Yes

4. Is the manuscript presented in an intelligible fashion and written in standard English?

Reviewer #1: Yes

Reviewer #2: Yes

5. Review Comments to the Author

Reviewer #1: This neatly organized and well written study summarizes the analyses of transplant outcomes in Korea over the last sixteen years. The manuscript is a useful presentation of transplantation experience in this country. Results on patient characteristics, immunosuppression therapy use and as wells graft rejection and loss statistics are presented. Overall, it is a good manuscript for readers wishing to learn about kidney transplantation system in Korea. Some important details were missed, however.

Most importantly, please kindly indicate the donor type in the patient cohort. Were all patients included in the study a deceased donor transplant recipients, or it was a mix of deceased and living donor transplant recipients. If the latter is the case, donor type (and donor relationship for living donors) should be shown in Tables 1, 3 and 4, and, importantly, included in Cox regression model as a variable and shown in Table 5. This would constitute a major revision to the manuscript.

Overall, I would caution against over-interpreting the results of variable distribution comparison between patients with or without acute rejection and graft loss, and Cox regression, due to the lack of information on donor type, ABO compatibility (whether or not ABOi transplants were included), pre-transplant sensitization level. The conclusions of the paper regarding risk factors impacting transplantation outcomes in Korea should be at least paraphrased to reflect these limitations. In addition, for a truly vigorous statistical testing of graft failure risk in a Cox regression model, testing for proportional hazard distribution of included variables would be beneficial, though I consider it optional for the scope of this manuscript.

Reviewer #2: The manuscript on the Outcomes of Kidney Transplantation Over a 16-year Period in Korea: An Analysis of

the National Health Information Database was read with great interest. Unfortunately the data presentation is not robust, and in-depth. The results must be reported mirroring the standard reporting of transplant data accepted universally for example 'survival free rejection' is no commonly used. Overall survival vs graft survival, % acute rejection (report Tcell vs AMR vs Mixed), what % was deceased donor vs Living Donor kidney tx., HLA match, ABO vs ABOi, Desensitization protocol used, ATG dosing?.

It is particularly interesting and counter to accepted results that ATG use led to worse outcomes. The authors do not explain this finding. The thesis was mixed, a more descriptive manuscript would suffice with the data and findings clearly presented.

A detailed table of results and a KM curve to present the results will be important. The discussion of the manuscript are centered around results not presented.

6. PLOS authors have the option to publish the peer review history of their article (what does this mean?). If published, this will include your full peer review and any attached files.

Reviewer #1: **Yes: **Dulat Bekbolsynov

Reviewer #2: No

---

## [Author Response · Author response to Decision Letter 0]

3 Feb 2021

Authors’ responses to the reviewers’ comments

First of all, we greatly appreciate your favorable review of our manuscript. In addition, we would also like to thank you for your valuable comments and suggestions. In response to the points that you raised, we would like to offer the following answers.

Reviewers’ comments

The manuscript is the summary in the detail analysis of transplant outcomes in Korea during the last sixteen years. The study include data about: patient characteristics; immunosuppression therapy; incidence of graft rejection and graft loss. Although, it is a useful review about the kidney transplantation system in Korea, the authors need to address all specifically indicated aspects by the reviewers.

The authors need to show the type of donors. It is also necessary to clearly show the results with deceased donors vs. live donors. Cosequently, tables 1, 3 and 4 need to be modified to clearly show the results for deceased donors and for live donors. In addition, the appropiate statistical analysis needs to be performed. All these changes will require significant revisions to the manuscript.

Response: We greatly appreciate your valuable opinions on our manuscript. We have added the results on the analysis of donor types and have revised Tables 1, 3, 4, and 5. We also have added descriptions regarding results of additional analyses to the revised manuscript page 5, 7, 11, 12, 14, and 17. We also have revised Fig 2 and 3.

The authors need to provide enough information supported by the appropiate statistical analysis to make sure that there is not an over-interpratation of their results. 

The authors need to address specific comments by the second reviewer: ABO compatibility (whether or not ABOi transplants were included), and pre-transplant sensitization levels. 

The conclusions need to be re-evaluated regarding risk factors impacting transplantation outcomes in Korea. 

Very careful ststistical analysis for graft failure risk needs to be performed using a Cox regression model, testing for proportional hazard distribution with variables.

Response: We agree with the opinion on the necessity of analyzing clinical data including ABOi and pre-transplant sensitization level. Unfortunately, it is difficult to investigate detailed clinical information, because we analyzed data from the database of the National Health Insurance claims. We have added some descriptions as the limitations of our study, and also have revised conclusions of our study (page 20-21) to reflect the reviewer’s suggestion. We also have reanalyzed the Cox’s regression by adding the donor types as a variable (Table 5). Cox regression model showed that the recent KT (during 2010-2017) have 9.732 hazard ratio when compared to preceding period (during 2002-2009). We also have added Fig 3B which reflects grafts survival according to the periods of KT to check and visualize the results of survival analyses.

In response to the reviewer # 1:

The presented data are not robust, and not in-depth. Additional data need to be included divided into deceased and living donors. The results need to reflect the standard reporting of transplant data accepted universally for example 'survival free rejection' is no commonly used. Overall survival vs graft survival, % acute rejection (report Tcell vs AMR vs Mixed), what % was deceased donor vs Living Donor kidney tx., HLA match, ABO vs ABOi, Desensitization protocol used, ATG dosing?. It is particularly interesting and counter to accepted results that ATG use led to worse outcomes. The authors do not explain this finding. The thesis was mixed, a more descriptive manuscript would suffice with the data and findings clearly presented.

Response: We have added the results on the analysis of donor types and have revised Tables 1, 3, 4, and 5, and the results of additional analyses have been added to the revised manuscript (page 5, 7, 11, 12, 14, and 17). We also have revised Fig 2 and 3. We agree with the reviewer’s opinion on the necessity of analyzing detailed clinical data including HLA match, ABO vs ABOi, desensitization protocol used. Because we analyzed data from the database of the National Health Insurance claims, it is difficult to investigate detailed clinical information and we have described these as the limitations of our study (page 20-21). In the current study, the use of ATG was analyzed as a variable which can be associated with for graft failure. Several recent studies have reported that the ATG induction show superiority over basiliximab induction for the prevention of acute rejection after KT. However, since the current study is a retrospective cohort study, it is not plausible to interpret that the use of ATG is direct cause of graft failure. It seems appropriate to interpret our results as a temporal relationship. We have added some descriptions on the interpretation of our data to the discussion section (page 19).

In response to Reviewer # 2:

This neatly organized and well written study summarizes the analyses of transplant outcomes in Korea over the last sixteen years. The manuscript is a useful presentation of transplantation experience in this country. Results on patient characteristics, immunosuppression therapy use and as wells graft rejection and loss statistics are presented. Overall, it is a good manuscript for readers wishing to learn about kidney transplantation system in Korea. Some important details were missed, however. Most importantly, please kindly indicate the donor type in the patient cohort. Were all patients included in the study a deceased donor transplant recipients, or it was a mix of deceased and living donor transplant recipients. If the latter is the case, donor type (and donor relationship for living donors) should be shown in Tables 1, 3 and 4, and, importantly, included in Cox regression model as a variable and shown in Table 5. This would constitute a major revision to the manuscript.

Response: We greatly appreciate your valuable opinions on our manuscript. We have added the results on the analysis of donor types and have revised Tables 1, 3, 4, and 5. We also have added descriptions regarding results of additional analyses to the revised manuscript page 5, 7, 11, 12, 14, and 17. We also have revised Fig 2 and 3.

Overall, I would caution against over-interpreting the results of variable distribution comparison between patients with or without acute rejection and graft loss, and Cox regression, due to the lack of information on donor type, ABO compatibility (whether or not ABOi transplants were included), pre-transplant sensitization level. The conclusions of the paper regarding risk factors impacting transplantation outcomes in Korea should be at least paraphrased to reflect these limitations. In addition, for a truly vigorous statistical testing of graft failure risk in a Cox regression model, testing for proportional hazard distribution of included variables would be beneficial, though I consider it optional for the scope of this manuscript.

Response: We agree with the reviewer’s opinion on the necessity of analyzing clinical data including ABOi and pre-transplant sensitization level. Unfortunately, it is difficult to investigate detailed clinical information, because we analyzed data from the database of the National Health Insurance claims. We have added some descriptions as the limitations of our study, and also have revised conclusions of our study (page 20-21) to reflect the reviewer’s suggestion. We also have reanalyzed the Cox’s regression by adding the donor types as a variable (Table 5). Cox regression model showed that the recent KT (during 2010-2017) have 9.732 hazard ratio when compared to preceding period (during 2002-2009). We also have added Fig 3B which reflects grafts survival according to the periods of KT to check and visualize the results of survival analyses.

---

## [Editor Report · Decision Letter 1]

8 Feb 2021

Outcomes of kidney transplantation over a 16-year period in Korea: An analysis of the National Health Information Database

PONE-D-20-37168R1

Dear Dr. Park,

We’re pleased to inform you that your manuscript has been judged scientifically suitable for publication and will be formally accepted for publication once it meets all outstanding technical requirements.

Kind regards,

Stanislaw Stepkowski

Academic Editor

PLOS ONE

Additional Editor Comments (optional):

None
---

## [Editor Report · Acceptance letter]

10 Feb 2021

PONE-D-20-37168R1 

Outcomes of kidney transplantation over a 16-year period in Korea: An analysis of the National Health Information Database 

Dear Dr. Park:

I'm pleased to inform you that your manuscript has been deemed suitable for publication in PLOS ONE. Congratulations! Your manuscript is now with our production department. 

Kind regards, 

on behalf of

Dr. Stanislaw Stepkowski 

Academic Editor

PLOS ONE